# FORKED DIFFUSION FOR CONDITIONAL GRAPH GENERATION

## ABSTRACT

We introduce a novel score-based diffusion framework that incorporates *forking* for conditional generation tasks. Specifically, a single *parent* diffusion process is associated with a primary variable (e.g., graph structure), while multiple *child* diffusion processes are employed, each dedicated to a dependent variable (e.g., graph property or label). The parent process guides the co-evolution of its child processes. Orchestrating, and regulating, the conditional flows effectively, this approach allows us to uncover intricate interactions and dependencies, and unlock new generative capabilities. We provide extensive experiments to demonstrate performance gains of the proposed method over contemporary baselines in the context of conditional graph generation, underscoring the potential of forking in challenging generative tasks such as inverse molecular design.

## 1 INTRODUCTION

The generation of high-quality data samples with desired properties is a fundamental challenge in numerous real-world applications, such as drug design, material synthesis, and image editing (Kotsias et al., 2020; Gebauer et al., 2022; Stokes et al., 2020; Meng et al., 2022; Couairon et al., 2023; Lugmayr et al., 2022). However, searching for new molecules with desired physicochemical properties (e.g., in a vast combinatorial space for molecules) makes it challenging for traditional brute-force methods to capture influential data factors. As a result, learning informative conditional representations is increasingly becoming imperative for tasks such as designing constrained molecules (Gebauer et al., 2019).

Deep generative models (DGMs) have shown remarkable ability to generate realistic data samples by accurately learning the underlying data distribution and mimicking the true generation process. In particular, these models have been deployed in various settings that involve conditional generation of high-quality data samples with desired properties. Among many DGMs that have gained widespread attention over recent years, Variational Autoencoders (VAEs), Generative Adversarial Networks (GANs), and Normalizing Flows (NFs) have found frequent use for both images (Higgins et al., 2017; Karras et al., 2020; Chen et al., 2016; Rezende & Mohamed, 2015) and graphs (Simonovsky & Komodakis, 2018; Lim et al., 2018; Jin et al., 2018; De Cao & Kipf, 2018; Verma et al., 2022).

Score-based diffusion models (SGMs) (Song et al., 2021) and probabilistic denoising diffusion models (DDPM) (Ho et al., 2020) have demonstrated superior capabilities in the estimation of complex data distributions in recent years. This class of models inject controlled noise gradually into the data through a series of small Markov steps during inference, resulting in the intentional degradation of information. Subsequently, these models are trained to remove the noise from the corrupted samples, effectively transforming them back into faithful data samples.

Diffusion offers a flexible training mechanism that can be extended to accommodate the requirements of downstream tasks. In particular, few mechanisms have been proposed to enable conditional data generation. Primarily, they fall in two categories: classifier-free (Ho & Salimans, 2022) and classifier-based (Dhariwal & Nichol, 2021) guidance. These techniques have been employed for tasks such as conditional image generation and text-to-image generation (Radford et al., 2021).

Diffusion techniques have also been adapted for graph-based applications. For instance, the denoising score matching mechanism introduced by Song et al. (2021) has found utility in diverse graph-related tasks. Specifically, Jo et al. (2022) have utilized this mechanism for unconditional

Table 1: Comparison of generative modeling methodologies.

| Method | Conditional | Energy-Guidance | Non-Autoregressive | Continuous-time | End-to-End | Authors |
|---|---|---|---|---|---|---|
| JT-VAE | ✗ | ✗ | ✗ | ✗ | ✓ | Jin et al. (2018) |
| ModFlow | ✗ | ✗ | ✓ | ✓ | ✓ | Verma et al. (2022) |
| GDSS | ✗ | ✗ | ✓ | ✓ | ✓ | Jo et al. (2022) |
| EDM | ✓ | ✗ | ✓ | ✗ | ✓ | Hoogeboom et al. (2022) |
| cG-SchNet | ✓ | ✗ | ✗ | ✗ | ✓ | Gebauer et al. (2022) |
| EEGSDE | ✓ | ✓ | ✓ | ✓ | ✗ | Bao et al. (2023) |
| JODO | ✓ | ✗ | ✓ | ✓ | ✓ | Huang et al. (2023) |
| **FDP** | ✓ | ✓ | ✓ | ✓ | ✓ | This work |

graph generation, while Corso et al. (2022) have applied it to computational molecular docking. Similarly, diffusion approaches have also been employed for conditional molecule generation. Initially, Hoogeboom et al. (2022) proposed an equivariant approach based on probabilistic denoising diffusion models (DDPM). This method has since been extended by Bao et al. (2023), who incorporated energy guidance into the framework, and more recently by Huang et al. (2023), who introduced further improvements by incorporating score-based modeling techniques.

In contrast to prior studies on diffusion, as outlined in Table 1, we introduce a novel approach to model conditional graphs. Our approach reimagines the diffusion process by postulating that structural evolution should occur in conjunction with the evolution of specific properties. Our objective in developing an alternative to existing conditional diffusion processes is to empower a model by bestowing it with a finer control over two key aspects: the 1) evolution of structural graph components, including nodes and edges, and 2) co-evolution of the structure in conjunction with one or more associated properties.

Toward that end, we define a novel conditional diffusion mechanism (please see Figure 1 for overview), **F**orked **D**iffusion **P**rocesses (**FDP**), which resembles *forking*. Specifically it defines a *parent* process over a primary variable and multiple independent *child* processes one per dependent variable. The parent process is able to guide the co-evolution of child processes, resulting in the learning of more flexible representations, which enable as well as benefit from the unraveling of complex interactions and dependencies. We establish rigorous theoretical underpinnings of our model by appealing to the theory of denoising score matching Song et al. (2021) and the tools from stochastic differential equations (SDEs) (Anderson, 1982), formalizing the validity of forked processes.

We demonstrate the merits of the proposed method with detailed empirical investigations on several standard constrained generation tasks with molecular and generic graph datasets, comparing with the latest advances in the field of conditional graph diffusion.

## 1.1 CONTRIBUTIONS

We now summarize our key contributions. We

- **(Conceptual and methodological)** introduce forking as a new technique for conditional generation, and propose an effective score-based, end-to-end trainable, non-autoregressive generative model designed for acquiring conditional representations.

  Our novel approach enables precise energy guidance through multiple property-conditioned forked diffusion processes.

- **(Technical)** provide a rigorous mathematical framework leveraging tools from Stochastic Differential Equations (SDEs) to derive both the forward forking diffusion process as well as the corresponding reverse SDE; and extend the formalism to incorporate additional contexts as conditioning information.

- **(Empirical)** demonstrate the versatility of the proposed forked diffusion mechanism (FDP) with strong empirical evidence. Our extensive evaluations showcase the superlative performance of the proposed model in diverse and challenging conditional graph generation settings across multiple datasets, surpassing contemporary baselines.

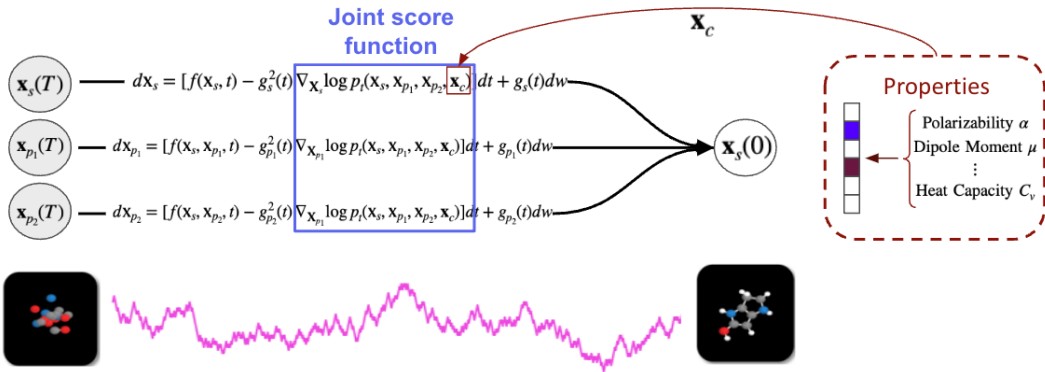

Figure 1: **Overview of the reverse process of forked diffusion process (FDP)**. We define the parent process over the primary variable i.e. structure $x_s$ and multiple forked child processes, one for each dependent variable, namely, $x_{p_1}$ and $x_{p_2}$. The variables start from a prior distribution $\mathcal{N}(0, I)$ and are guided via a joint score function with an additional context $\mathbf{x}_c$ to generate the data-sample conditioned on this context.

## 2 RELATED WORKS

**Diffusion guidance**   Guidance is a technique applied to regulate the diffusion process for conditional generation. Previous approaches, such as those by Dhariwal & Nichol (2021) and Song et al. (2021), employed classifier-based guidance for class-conditional image generation. This concept was extended to text-to-image generation using CLIP (Radford et al., 2021). Recently, Bao et al. (2023) introduced a method that trains an energy guidance property classifier on top of a pre-trained diffusion model to enhance the generation of molecular graphs, akin to classifier guidance. Unlike prior approaches, we introduce a novel strategy for guidance by 1) learning *forked* diffusion flows during training, where conditioning is requested, and 2) employing multiple interacting processes during sampling. Notably, our method eliminates the need for training a separate classifier, a feature referred to as "End-to-End" in Table 1, distinguishing it from methodologies such as (Dhariwal & Nichol, 2021) for images and (Bao et al., 2023) for graphs.

**Conditional Diffusion for Graphs**   Recent advances in generative modeling have leveraged score-based techniques employing diffusion or stochastic differential equations Jo et al. (2022); Liu et al. (2021); Jing et al. (2022); Guth et al. (2022); Ingraham et al. (2022). These approaches have been extended to the realm of conditional generation (Hoogeboom et al., 2022; Gebauer et al., 2022; 2019; Bao et al., 2023; Verma et al., 2023), wherein pre-trained generative models are fine-tuned to produce 3D molecules conditioned on specific properties. EDM (Hoogeboom et al., 2022) introduces a DDPM (Ho et al., 2020) designed for conditional generation by incorporating properties into node features during training. More recently, (Huang et al., 2023) introduced JODO, a score-based diffusion model for conditional generation, demonstrating performance improvements over EDM (Hoogeboom et al., 2022) and EEGSDE (Bao et al., 2023). Our method is also inspired by stochastic differential equations (SDE), similar to GDSS (Jo et al., 2022) and JODO (Huang et al., 2023), where we aim to learn more expressive representation by having interacting and separated forked diffusion processes based on properties, aimed at conditional generation.

## 3 FORKED DIFFUSION PROCESSES

Diffusion models present a versatile recipe for the construction of stochastic processes, playing a crucial role in generative modeling (Jo et al., 2022; Song et al., 2021) and decision-making (Ajay et al., 2023). In this work, we appeal to the stochastic differential equation (SDE)-based diffusion framework as proposed by Song et al. (2021) to define our forked diffusion process. The next subsections will delve into the mathematical details of our method.

## 3.1 Forked Diffusion Processes

We present a new paradigm for diffusion models, where we leverage the data structure to diffuse multiple processes and achieve a more robust representation. Specifically, we define a parent process over the primary variable $\mathbf{y}_s$ (e.g., structure) and a child process over each dependent variable $\mathbf{y}_i$ (e.g., property). We define the parent forward process over the primary variable $\mathbf{y}_s$ as,

$$d\mathbf{y}_s = \mathbf{f}_s(\mathbf{y}_{s,t}, t)dt + g_s(t)d\mathbf{w} \tag{1}$$

where $\mathbf{f}_s$ and $g_s$ are corresponding diffusion and drift functions, and $d\mathbf{w}$ is the weiner noise. Now, we define the child forward process over $k$ dependent variables $\mathbf{y} = \{\mathbf{y}_1, \ldots, \mathbf{y}_k\}$ as,

$$d\mathbf{y}(t) = \begin{pmatrix} d\mathbf{y}_1(t) \\ \vdots \\ d\mathbf{y}_k(t) \end{pmatrix} = \begin{pmatrix} \mathbf{f}_p(\mathbf{y}_{1,t}, \mathbf{y}_{s,t}, t)dt + g_p(t)d\mathbf{w} \\ \vdots \\ \mathbf{f}_p(\mathbf{y}_{k,t}, \mathbf{y}_{s,t}, t)dt + g_p(t)d\mathbf{w} \end{pmatrix} \tag{2}$$

Here, $\mathbf{f}_p$ and $g_p$ denote the diffusion and drift functions, respectively, corresponding to $k$ child processes related to dependent variables. Collectively, along with the parent forward process, they constitute our forked diffusion process. Like diffusion processes (Song et al., 2021), these operations introduce random Gaussian noise to the data to approach a prior or uninformed distribution. We will now outline the reverse diffusion process.

## 3.2 Reverse Diffusion Process

The reverse diffusion process can be simulated by starting from a prior distribution and moving towards the data point, given by the reverse SDE as:

$$d\mathbf{x} = [f(\mathbf{x}, t) - g_t^2 \nabla_x \log p_t(\mathbf{x})]dt + g_t d\bar{\mathbf{w}} \tag{3}$$

The term $\nabla_x \log p_t(\mathbf{x})$ resembles the score function, responsible for guiding the diffusion process that generates data from the prior distribution, with its parameters defined by a neural network. We aim to integrate scores from our child-forked process to guide the parent diffusion process in generating primary variables. We exploit the independence among the *child* variables in the forking mechanism to modify the reverse process as,

> **Proposition 1: Reverse SDE without context**
>
> The reverse SDE for the forward SDE system $\mathbf{y}_t = \{\mathbf{y}_{s,t}, \mathbf{y}_{1,t}, \ldots, \mathbf{y}_{k,t}\}$, when considering the conditional dependence on dependent variables $\{\mathbf{y}_1, \ldots, \mathbf{y}_k\}$ is shown in Eq. 4. For more details and derivation, see the Appendix A.1.
>
> $$d\mathbf{y}_t = [\mathbf{f}(\mathbf{y}_t, t) - g_t^2 \nabla_{\mathbf{y}_t} \log p_t(\mathbf{y}_t)]dt + g_t d\bar{\mathbf{w}} \tag{4}$$

We exploit the independence among dependent variables $\{\mathbf{y}_1, \ldots, \mathbf{y}_k\}$, to factorize the score function as,

$$p_t(\mathbf{y}_{s,t}, \mathbf{y}_{1,t}, \ldots, \mathbf{y}_{k,t}) = p_t(\mathbf{y}_{1,t}, \ldots, \mathbf{y}_{k,t} \mid \mathbf{y}_{s,t}) p_t(\mathbf{y}_{s,t}) = \prod_i^k p_t(\mathbf{y}_{i,t} \mid \mathbf{y}_{s,t}) p_t(\mathbf{y}_{s,t})$$

$$\tag{5}$$

$$\nabla_{\mathbf{y}_t} \log p_t(\mathbf{y}_{s,t}, \mathbf{y}_{1,t}, \ldots, \mathbf{y}_{k,t}) = \nabla_{\mathbf{y}_t} \log p_t(\mathbf{y}_{s,t}) + \sum_i^k \nabla_{\mathbf{y}_t} \log p_t(\mathbf{y}_{i,t} \mid \mathbf{y}_{s,t})$$

Plugging it into the main Eq. 4, we obtain,

$$d\mathbf{y}_t = [\mathbf{f}(\mathbf{y}_t, t) - g_t^2 (\nabla_{\mathbf{y}_t} \log p_t(\mathbf{y}_{s,t}) + \sum_i^k \nabla_{\mathbf{y}_t} \log p_t(\mathbf{y}_{i,t} \mid \mathbf{y}_{s,t}))]dt + g_t d\bar{\mathbf{w}} \tag{6}$$

## 3.3 Conditional score function

We expand our proposed approach to enable conditional generation with an external context $\mathbf{y}_C = \{\mathbf{y}_c \mid c \in C\}$, where $C \subseteq \{1, \ldots, k\}$. This context can be represented as a scalar or vector, describing a particular value associated with a data-dependent variable. For example, it could

Table 2: **FDP** comparison to Classifier-based (Dhariwal & Nichol, 2021) and Classifier-free guidance (Ho & Salimans, 2022) applied for conditional generation in Diffusion models. Here, the $\mathbf{x}$ represents the primary data variable, and $y$ is a scalar value which is the external context for conditional generation, the $f_\phi$ is the classifier trained for classifier-based guidance, and $\epsilon_\theta$ is a learnable score model parametrized by $\theta$. For **FDP**, we have one parent and child process where $\mathbf{x}$ represents the primary data variable, $y$ is the external context for conditional generation, and $\mathbf{y}'$ is the only child process forked from $\mathbf{x}$, i.e., for the same property as $y$, $s_{\theta,\phi}$ are learnable score models.

| Method | Diffusion Scheme | Approach |
|---|---|---|
| **Classifier-based** | $d\mathbf{x} = \mathbf{f}(\mathbf{x},t)dt + g(t)d\mathbf{w}$ 
 $d\mathbf{x} = [f(\mathbf{x},t) - g_t^2 \nabla_x \log p_t(\mathbf{x},y)]dt + g_t d\bar{\mathbf{w}}$ | $\nabla_{\mathbf{x}_t} \log p(\mathbf{x}_t, y) = \nabla_{\mathbf{x}_t} \log p(\mathbf{x}_t) + \nabla_{\mathbf{x}_t} \log p(y \mid \mathbf{x}_t)$ 
 $\approx -\frac{1}{\sqrt{1-\bar{\alpha}_t}} \epsilon_\theta(\mathbf{x}_t, t) + \nabla_{\mathbf{x}_t} \log f_\phi(y \mid \mathbf{x}_t)$ |
| **Classifier-free** | $d\mathbf{x} = \mathbf{f}(\mathbf{x},t)dt + g(t)d\mathbf{w}$ 
 $d\mathbf{x} = [f(\mathbf{x},t) - g_t^2 \nabla_x \log p_t(\mathbf{x},y)]dt + g_t d\bar{\mathbf{w}}$ | $\nabla_{\mathbf{x}_t} \log p(y \mid \mathbf{x}_t) = \nabla_{\mathbf{x}_t} \log p(\mathbf{x}_t \mid y) - \nabla_{\mathbf{x}_t} \log p(\mathbf{x}_t)$ 
 $= -\frac{1}{\sqrt{1-\bar{\alpha}_t}} (\epsilon_\theta(\mathbf{x}_t, y) - \epsilon_\theta(\mathbf{x}_t, t))$ |
| **FDP** | $d\mathbf{x} = \mathbf{f}(\mathbf{x},t)dt + g(t)d\mathbf{w},\ d\mathbf{y}' = \mathbf{f}_y(\mathbf{x},\mathbf{y}',t)dt + g_y(t)d\mathbf{w}$ 
 $d\mathbf{x} = [f(\mathbf{x},t) - g_t^2 \nabla_x \log p_t(\mathbf{x},y,\mathbf{y}')]dt + g_t d\bar{\mathbf{w}}$ 
 $d\mathbf{y}' = [f(\mathbf{x},\mathbf{y}',t) - g_t^2 \nabla_{\mathbf{y}'} \log p_t(\mathbf{x},y,\mathbf{y}')]dt + g_t d\bar{\mathbf{w}}$ | $\nabla \log p_t(\mathbf{x}_t, y, \mathbf{y}') = \nabla \log p_t(\mathbf{x}_t, y) + \nabla \log p_t(\mathbf{y}' \mid \mathbf{x}_t, y)$ 
 $\nabla \log p_t(\mathbf{x}_t, y) \approx s_{\theta,t}(\mathbf{x}_t, y),\ \nabla \log p_t(\mathbf{y}' \mid \mathbf{x}_t, y) \approx s_{\phi,t}(\mathbf{x}, y, \mathbf{y}')$ 
 $\nabla \log p_t(\mathbf{x}_t, y, \mathbf{y}') = s_{\theta,t}(\mathbf{x}_t, y) + s_{\phi,t}(\mathbf{x}, y, \mathbf{y}')$ |

represent properties such as Synthetic Accessibility (SA) score or plogp in the case of molecules or image labels for images. This extension modifies the joint distribution for the score function in Equation 5 as follows:

---

**Proposition 2: Reverse SDE with context**

The reverse SDE for the system $\mathbf{y}_t = \{\mathbf{y}_{s,t}, \mathbf{y}_{1,t}, \ldots, \mathbf{y}_{k,t}\}$ provided an external conditioning context $\mathbf{y}_C$, is shown in Eq. 7. The score function $\nabla_{\mathbf{y}_t} \log p_t(\mathbf{y}_t, \mathbf{y}_C)$ can be factorized as shown in Eq. 8, leading to factorized parameterization in Eq. 8. For more details, see Appendix A.2.

$$\mathrm{d}\mathbf{y}_t = [\mathbf{f}(\mathbf{y}_t, t) - g_t^2 \nabla_{\mathbf{y}_t} \log p_t(\mathbf{y}_t, \mathbf{y}_C)]\mathrm{d}t + g_t \mathrm{d}\bar{\mathbf{w}} \tag{7}$$

---

We utilize our independence assumption to factorize the distribution as where the external context $\mathbf{y}_C$ affects the process corresponding to its child and parent processes, thus remaining independent of other child processes.

$$p_t(\mathbf{y}_{s,t}, \mathbf{y}_{1,t}, \ldots, \mathbf{y}_{k,t}, \mathbf{y}_C) = \prod_i^k p_t(\mathbf{y}_{i,t} \mid \mathbf{y}_{s,t}, \mathbf{y}_C) p_t(\mathbf{y}_{s,t}, \mathbf{y}_C) \tag{8}$$

Given the distribution obtained in Equation 8, the score function factorizes as follows:

$$\begin{aligned}
\nabla_{\mathbf{y}_t} \log p_t(\mathbf{y}_{s,t}, \mathbf{y}_{1,t}, \ldots, \mathbf{y}_{k,t}, \mathbf{y}_C) = {} & \nabla_{\mathbf{y}_t} \log p_t(\mathbf{y}_{s,t}, \mathbf{y}_C) \\
& + \sum_{i \notin C}^k \nabla_{\mathbf{y}_t} \log p_t(\mathbf{y}_{i,t} \mid \mathbf{y}_{s,t}) \\
& + \sum_c^C \sum_i^k \delta_{i=c} \nabla_{\mathbf{y}_t} \log p_t(\mathbf{y}_{i,t} \mid \mathbf{y}_{s,t}, \mathbf{y}_c)
\end{aligned} \tag{9}$$

The conditional reverse SDE is obtained by plugging the score from Eq 9 into Eq. 7. Our method offers a novel approach to integrating external contextual information into conditional generation. We compare our approach with conventional classifier-based and classifier-free guidance methods in Table 2. Given a trained conditional model, our generative process begins by sampling an external context or conditioning value $\mathbf{y}_C$, which can also be supplied externally. We then simulate the reverse diffusion process, similar to the one described in Equations 9 and 7, but with a modified score function to generate the data.

### 3.4 TRAINING OBJECTIVE

The score functions can be estimated by training the time-dependent score-based models $s_{\theta,t}$ and $s_{\phi_i,t}$ such that they approximate the score-matching objective (Hyvärinen & Dayan, 2005). Our

---

**Algorithm 1** Training FDP

**Require:** Dataset $\mathcal{D}$, iterations $n_{\text{iter}}$, batch size $B$, number of batches $n_B$, $K$ properties to consider
1: Initialise parameters $s_{\theta,t}, \{s_{\phi_i,t}\}_{i=1}^K$ for Score Networks
2: **for** $k = 1, \ldots, n_{\text{iter}}$ **do**
3:    **for** $b = 1, \ldots, n_B$ **do**
4:       $t \sim \mathcal{U}(0, 1]$
5:       $\mathcal{D}_b = \{(\boldsymbol{y}_{s,l}, \{\boldsymbol{y}_{i,l}\}_{i=1}^K)_{l=1}^B, \boldsymbol{y}_C\} \sim \mathcal{D}$
6:       $\mathcal{L}_b \leftarrow$ Eq. 10
7:    **end for**
8:    $\theta, \{\phi_i\}_{i=i}^K \leftarrow \texttt{optim}(\frac{1}{n_B}\sum_{b=1}^{n_B} \mathcal{L}_b)$
9: **end for**

---

**Algorithm 2** Generating with FDP

**Require:** Score-based models $s_{\theta,t}, \{s_{\phi_i,t}\}_{i=1}^K$, Time step schedule $\{t\}_{t=T}^0$, Langevin MCMC step size $\alpha$, External context $\boldsymbol{y}_C$
1: $\boldsymbol{y}_{s_T}, \{\boldsymbol{y}_{i,T}\}_{i=1}^K \sim \mathcal{N}(0, I)$
2: **for** $t = T, \ldots, 0$ **do**
3:    $s_{\theta,t} \leftarrow s_{\theta,t}(\boldsymbol{y}_{s_t}, \{\boldsymbol{y}_{i,t}\}_{i=1}^K, \boldsymbol{y}_C)$
4:    $\{s_{\phi_i,t}\}_{i=1}^K \leftarrow \{s_{\phi_i,t}(\boldsymbol{y}_{s_t}, \boldsymbol{y}_{i,t}, \boldsymbol{y}_C)\}_{i=1}^K$
5:    $\boldsymbol{y}_{s_t} \leftarrow \boldsymbol{y}_{s_t} + \frac{\alpha}{2}s_{\theta,t} + \sqrt{\alpha}z_s; z_s \sim \mathcal{N}(0, I)$
6:    $\boldsymbol{y}_{i_t} \leftarrow \boldsymbol{y}_{i_t} + \frac{\alpha}{2}s_{\phi_i,t} + \sqrt{\alpha}z_i; z_i \sim \mathcal{N}(0, I)$
7: **end for**

---

training routine, reported in Algorithm 1 leverages a denoising score matching technique (Song et al., 2021; Jo et al., 2022). Specifically, the distribution $p_{0t}(\mathbf{y}_t | \mathbf{y}_0)$ is used for updating from $p_0$ to $p_t$ during the forward diffusion, and for sampling both $\mathbf{y}_0 \sim p_{\text{data}}$ and $\mathbf{y}_t \sim p_{0t}(\mathbf{y}_t | \mathbf{y}_0)$, where $\mathbf{y}_t = \{\mathbf{y}_{s,t}, \{\mathbf{y}_{i,t}\}_{i=1}^K\}$ and $\mathbf{y}_C$ is the external contextual information. As a result, we provide an objective function for optimizing the score networks $\boldsymbol{s}_\theta, \boldsymbol{s}_{\phi_i}$, which we write as follows:

$$\min_{\theta,\phi_i} \mathbb{E}_t \left\{ \lambda_{\mathbf{y}_t}(t)\mathbb{E}_{\mathbf{y}_0}\mathbb{E}_{\mathbf{y}_t|\mathbf{y}_0} \| \boldsymbol{s}_{\theta,t}(\mathbf{y}_{s,t}, \mathbf{y}_c) + \sum_i^k \boldsymbol{s}_{\phi_i,t}(\mathbf{y}_{i,t}, \mathbf{y}_{s,t}, \mathbf{y}_c) - \nabla_{\mathbf{y}_t} \log p_{0t}(\mathbf{y}_t, \mathbf{y}_C) \|_2^2 \right\} \quad (10)$$

where $\mathbb{E}_{\mathbf{y}_0} = \mathbb{E}_{\mathbf{y}_{s,0}, \mathbf{y}_{i,0}}$ and $\mathbb{E}_{\mathbf{y}_t} = \mathbb{E}_{\mathbf{y}_{s,t}, \mathbf{y}_{i,t}}$. It is worth noting that the additional influence introduced by the variable $\boldsymbol{s}_{\phi_i}$ can be conceptually characterized as an *energy guidance* mechanism for the properties under consideration. This guidance mechanism operates in conjunction with the structural information provided by $\boldsymbol{s}_\theta$, resulting in a novel form of guidance that is orchestrated by a branching diffusion process.

## 4 EXPERIMENTS

We evaluate our model on the task of generating molecules conditioned on *chemical* properties in Section 4.1 and quantum properties in Section 4.2. We evaluate the performance of our method for conditional generic graph generation in Section 4.3 and for unconditional molecule generation in Section 4.4.

### 4.1 CONDITIONAL GENERATION ON MOLECULAR CHEMICAL PROPERTIES

**Task** In this experiment, the objective is to generate molecules while conditioning on specific chemical properties. We incorporate three key chemical properties for conditioning: Penalized logP (plogP), Quantitative Estimate of drug-likeness (QED), and SA score. These properties are calculated based on a given molecule using the RDKit library (Landrum et al., 2016). We consider QM9 and ZINC250k data Ramakrishnan et al. (2014); Irwin et al. (2012), following the evaluation setup from (Jo et al., 2022). More details on our model parameterization, forking setup, datasets are given in Section B.2.

**Baseline** To assess the effectiveness of the proposed forking methodology, we introduce a conditional version of GDSS (Jo et al., 2022) that relies on classifier-free guidance. This modified GDSS baseline model is capable of conditioning graph information on a single chemical property by incorporating the conditional property vector into the node feature matrix, thereby learning conditional representations. Specifically, the score network is conditioned as follows: $\boldsymbol{s}_\theta = \text{MLP}(\text{GNN}([\boldsymbol{X}, \mathbf{y}_c], \boldsymbol{A}))$, where $\boldsymbol{X}$ represents node features, $\boldsymbol{A}$ is the adjacency matrix, and $\mathbf{y}_c$ denotes the property being conditioned upon.

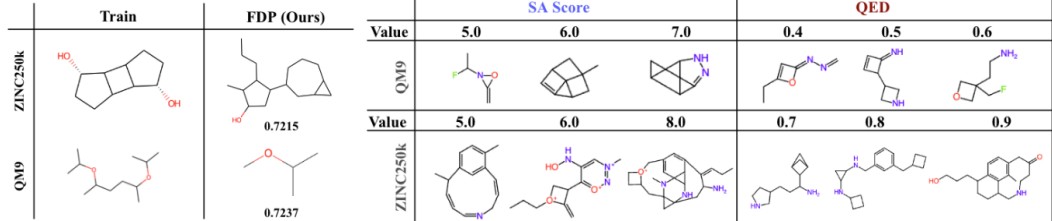

Figure 2: **Left**: Generated molecules and the corresponding Tanimoto similarity calculated concerning the dataset **Right**: **Visualization of Conditional Molecule generation**. We conditioned on property values (denoted as **Value** row), and the shown molecule has the true property value within two places after the decimal. The top row shows results from QM9, and the bottom row is from the ZINC250k dataset.

Table 3: **Left**: Molecular metrics on QM9. Our model outperforms GDSS on all metrics, obtaining lower FCD and MMD values while achieving higher Validity and Novelty. **Right**: MAE results for the QM9 and ZINC250k datasets. The forked diffusion process enables our model to obtain lower MAE values.

| Property | Method | Validity % ($\uparrow$) | FCD ($\downarrow$) | MMD ($\downarrow$) | Novelty % ($\uparrow$) | Prop | Method | ZINC MAE $\downarrow$ | QM9 MAE $\downarrow$ |
|---|---|---|---|---|---|---|---|---|---|
| plogp | GDSS | 38.4 | 17.63 | 0.120 | 100 | plogp | GDSS | 8.84 | 7.82 |
|  | **FDP** | **54.44** | **17.43** | **0.118** | 100 |  | **FDP** | **6.43** | **7.37** |
| QED | GDSS | 32.51 | 16.91 | 0.125 | 100 | QED | GDSS | 0.15 | 0.10 |
|  | **FDP** | **50.76** | **16.21** | 0.125 | 100 |  | **FDP** | **0.06** | **0.08** |
| SA Score | GDSS | 61.25 | 19.12 | 0.124 | 85 | SA score | GDSS | 6.07 | 9.08 |
|  | **FDP** | **88.87** | **18.17** | **0.119** | **100** |  | **FDP** | **5.64** | **8.45** |

**Metrics** First, we evaluate the performance for molecular generation via established metrics, including Validity, Novelty, Maximum Mean Discrepancy (MMD), and Fréchet ChemNet Distance (FCD). Second, we evaluate the effectiveness of conditioning by computing the Mean Absolute Error (MAE) between the intended ground truth property and the property extracted via RDKit from the molecule sampled via Algorithm 2.

**Results** In Table 3 (right), we present the MAE results for QM9 and ZINC250k datasets. The consistently lower MAE values achieved by our model, compared to GDSS, highlight its proficiency in generating molecules with the desired chemical properties. These results underscore the substantial improvements facilitated by the forking mechanism. In Table 3 (left), our model consistently outperforms GDSS on the QM9 dataset across various graph generation metrics, including Validity, Novelty, MMD, and FCD. Figure 2 illustrates the Tanimoto similarity scores obtained with our model and provides visual samples of molecules generated by conditioning on specific chemical property values.

## 4.2 CONDITIONAL GENERATION ON MOLECULAR QUANTUM PROPERTIES

**Task** In this experiment, we evaluate the effectiveness of the proposed forked diffusion mechanism in the domain of molecular generation, with a specific focus on conditioning the process on six quantum properties sourced from the QM9 dataset (Ramakrishnan et al., 2014). We follow the evaluation setup from (Huang et al., 2023), and implement the forking based on the Diffusion Graph Transformer (DGT) model. Further details on forking and parameterizations settings are provided in Section B.3.

**Metrics** In the evaluation, we follow previous works Hoogeboom et al. (2022); Huang et al. (2023) by using the MAE to measure the disparity between the ground truth properties and the properties of molecules generated by the model being conditioned on that property value.

**Results** We conducted a comprehensive comparison of our approach with several prominent conditional generative models, including JODO (Huang et al., 2023), EDM (Hoogeboom et al., 2022) and EEGSDE (Bao et al., 2023). The results of our evaluation are summarized in Table 4. It's evident from the findings that our model consistently outperforms these multiple conditional diffusion baselines, namely JODO, EDM, and EEGSDE, across all the evaluated quantum properties

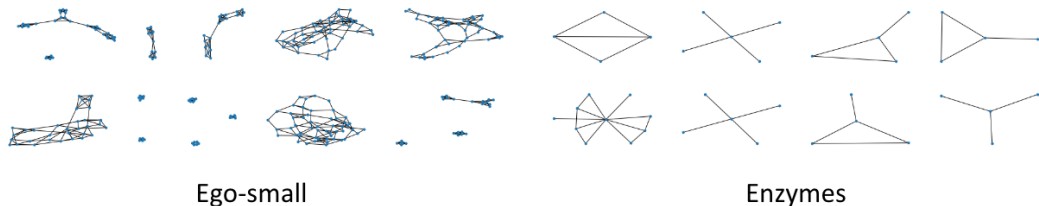

Ego-small                                                             Enzymes

Figure 3: Samples of graphs generated by **FDP** on Enzymes and Ego-small dataset.

Table 4: Evaluation for molecule generation conditioned on single target quantum property for the QM9 dataset. Our model consistently achieves lower MAE values across all the six evaluated properties.

| Method | $C_v$ ($\frac{cal}{mol}$K) | $\mu$ (D) | $\alpha$ (Bohr$^3$) | $\Delta\epsilon$ (meV) | $\epsilon_{HOMO}$ (meV) | $\epsilon_{LUMO}$ (meV) |
|---|---|---|---|---|---|---|
| U-bound | 6.879±0.015 | 1.613±0.003 | 8.98±0.02 | 1464±4 | 645±41 | 1457±5 |
| # Atoms | 1.971 | 1.053 | 3.86 | 866 | 426 | 813 |
| EDM | $1.065 \pm 0.010$ | $1.123 \pm 0.013$ | $2.78 \pm 0.04$ | $671 \pm 5$ | $371 \pm 2$ | $601 \pm 7$ |
| EEGSDE | $0.941 \pm 0.005$ | $0.777 \pm 0.007$ | $2.50 \pm 0.02$ | $487 \pm 3$ | $302 \pm 2$ | $447 \pm 6$ |
| JODO | $0.581 \pm 0.001$ | $0.628 \pm 0.003$ | $1.42 \pm 0.01$ | $335 \pm 3$ | $226 \pm 1$ | $256 \pm 1$ |
| FDP | $\mathbf{0.559} \pm 0.002$ | $\mathbf{0.627} \pm 0.001$ | $\mathbf{1.36} \pm 0.01$ | $\mathbf{323} \pm 2$ | $\mathbf{225} \pm 1$ | $\mathbf{244} \pm 3$ |
| L-bound | 0.040 | 0.043 | 0.09 | 65 | 39 | 36 |

Table 5: Evaluations on MMD statistics and MAE values on Ego-small and Enzyme datasets. **FDP** consistently outperforms the competing method across all the metrics for the considered properties.

| Property | Method | | Ego-small | | | | | Enzymes | | | |
|---|---|---|---|---|---|---|---|---|---|---|---|
| | | Degree ↓ | Clustering ↓ | Orbit ↓ | Spectral ↓ | MAE ↓ | Degree ↓ | Clustering ↓ | Orbit ↓ | Spectral ↓ | MAE ↓ |
| Nodes | GDSS | 12.7 | 25.4 | 0.6 | 3.8 | 3.0 | 50.2 | 21.5 | 16.0 | 0.7 | 16.4 |
| | **FDP** | **3.5** | **6.4** | **0.2** | **1.1** | **2.9** | **5.6** | **7.6** | **2.6** | **0.5** | **15.3** |
| Edges | GDSS | 10.9 | 21.7 | 0.5 | 2.8 | 6.1 | 19.5 | 10.7 | 2.5 | 0.5 | 23.8 |
| | **FDP** | **1.8** | **4.9** | **0.5** | **2.4** | **3.7** | **4.2** | **5.2** | **0.6** | 0.5 | **20.8** |
| Density | GDSS | 4.0 | 10.6 | 0.5 | 2.6 | 0.18 | 16.8 | 9.0 | 2.1 | 0.5 | 0.085 |
| | **FDP** | **0.6** | **1.6** | **0.2** | **1.1** | 0.18 | **2.6** | **6.7** | **1.2** | 0.5 | **0.081** |
| ANND | GDSS | 9.3 | 20.1 | 0.4 | 3.8 | 2.0 | 17.3 | 8.1 | 5.5 | 0.6 | 0.5 |
| | **FDP** | **2.3** | **3.6** | **0.2** | **1.1** | 2.0 | **10.1** | **7.3** | **1.5** | 0.6 | **0.4** |

within the QM9 dataset. The consistently superior performance of our model can be attributed to the effective conditional guidance facilitated by our forked diffusion mechanism.

## 4.3 CONDITIONAL GENERATION ON GENERIC GRAPH PROPERTIES

**Task** In this experiment, we employ the forked diffusion technique to assess its performance on generic graphs while conditioning on the following graph properties: (i) Number of nodes, (ii) Number of edges, (iii) Density and (iv) ANND: Average Nearest Neighbor Degree. Our evaluation setup follows (Jo et al., 2022), by evaluating on Ego-small and Enzymes datasets (Sen et al., 2008; Schomburg et al., 2004). We consider a single baseline which is the conditional GDSS model that we described in Section 4.1. Further details on models and dataset setups are given in Section B.4.

**Metrics** In terms of evaluation metric, we measure the MAE between the property values of the generated graphs (via the NetworkX library (Hagberg et al., 2008) and the ground truth conditional property. Furthermore, we evaluate standard MMD statistics (You et al., 2018; Jo et al., 2022), including Degree, Clustering Coefficient, Orbit Count, and Spectral MMD.

**Results** The results are displayed in Tables 5, and Figure 3 provides examples of generated graphs. Our observations reveal that **FDP** consistently outperforms the baseline methods across all evaluated properties. Notably, the lower MAE values attained by **FDP** compared to other methods highlight its exceptional capacity to finely control conditional information flow, resulting in superior sample generation.

Figure 4: Samples of molecules generated by our model for unconditional molecule generation.

**QM9**                **ZINC250k**

Table 6: Molecular graph metrics for unconditional generation on the QM9 and ZINC250k dataset. **FDP** outperforms current baselines in achieving the highest validity score.

| | QM9 | | | | ZINC250k | | | |
|---|---|---|---|---|---|---|---|---|
| | Validity % (↑) | MMD (↓) | FCD (↓) | Novelty % (↑) | Validity % (↑) | MMD (↓) | FCD (↓) | Novelty % (↑) |
| GraphAF+FC | 74.43 | 0.021 | 5.625 | 86.59 | 68.47 | 0.044 | **16.023** | 99 |
| GraphDF+FC | 93.88 | 0.064 | 10.928 | **98.54** | 90.61 | 0.177 | 33.546 | **100** |
| MoFlow | 91.36 | 0.017 | 4.467 | 94.72 | 63.11 | 0.046 | 20.931 | **100** |
| EDP-GNN | 47.52 | 0.005 | 2.680 | 86.58 | 82.97 | 0.049 | 16.737 | **100** |
| GraphEBM | 8.22 | 0.030 | 6.143 | 97.01 | 5.29 | 0.212 | 35.471 | **100** |
| GDSS-seq | 94.47 | 0.010 | 4.004 | 85.48 | 92.39 | **0.030** | 16.847 | **100** |
| **FDP** | **95.21** | **0.004** | **2.434** | 96.16 | **94.40** | 0.033 | 17.937 | **100** |

## 4.4 UNCONDITIONAL MOLECULE GENERATION

**Task** We assess the performance of our model for unconditional generation of molecular structures, on the QM9 and ZINC250K datasets (Ramakrishnan et al., 2014; Irwin et al., 2012). We compare our model to several contemporary autoregressive models, including GraphAF+FC (Shi et al., 2020) and GraphDF+FC (Luo et al., 2021), as well as one-shot methods, including MoFlow (Zang & Wang, 2020), GraphEBM (Liu et al., 2021), GDSS (Jo et al., 2022), following the experimental setup from Jo et al. (2022).

**Results** The results pertaining to molecular graph generation metrics have been systematically documented and are accessible in Table 6. This comprehensive analysis reveals that our model surpasses other baseline models in a majority of the test cases, demonstrating notable strengths in terms of Validity and Novelty. This performance superiority is consistent across both the QM9 and ZINC250k datasets. Visual representations of molecular structures generated by our model are provided for reference in Figure 4.

## 5 CONCLUSION, BROADER IMPACT, AND LIMITATIONS

We introduced forking as a novel approach to model conditional information within generative models tailored for graph data. FDP incorporates an effective mechanism to control the overall generative process: it bifurcates diffusion into a parent process and multiple child processes. Our experimental results showcase the superior performance of FDP when compared to current state-of-the-art baselines across various tasks. Specifically, FDP demonstrates exceptional capabilities in the domains of conditional graph generation for molecular structures, inverse molecule design tasks, and the generation of generic graphs, surpassing contemporary diffusion-based methods. These findings underscore the utility of the forking mechanism as an effective means of enhancing the accuracy of conditional predictions within generative models for graph data.

Conditional generation is fast emerging as one of the most exciting avenues within machine learning, and would benefit from techniques beyond classifier-based and classifier-free schemes. Thus, forking as a concept should be broadly applicable in conditional settings beyond this work. That said, the current work has focused solely on conditional graph generation, and the efficacy of forking in other domains (e.g, conditional image and text generation settings) needs to be investigated.

**Reproducibility statement** All the experiments used in our evaluation are reproducible, and upon acceptance we will make all our code and trained model available under the MIT License on Github.

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

# A    PROOFS

## A.1    DERIVATION OF THE REVERSE SDE

For a Stochastic Differential Equation (SDE) of the form,

$$dx = f(x_t, t)\mathrm{d}t + g(x_t, t)\mathrm{d}\mathbf{w} \tag{11}$$

where $f(\cdot)$ and $g(\cdot)$ are diffusion, drift function and $\mathrm{d}\mathbf{w}$ is the weiner noise. The evolution of the distribution of $x_t$ is governed by the Kolmogorov Forward Equation (KFE) as,

$$\partial_t p\left(x_t\right) = -\partial_{x_t}\left[f\left(x_t\right)p\left(x_t\right)\right] + \frac{1}{2}\partial_{x_t}^2\left[g^2\left(x_t\right)p\left(x_t\right)\right] \tag{12}$$

**Kolmogrov Forward/Backward Equation (KFE/KBE)**    Essentially KFE describes the evolution of a probability distribution $p(x_t)$ forward in time. The reverse-time SDE can be derived by solving the Kolmogorov Backward Equation (K.B.E) as derived in Anderson (1982). It can be defined for $t_1 \geq t_0$ as,

$$-\partial_t p\left(x_{t_1} \mid x_{t_0}\right) = f\left(x_{t_0}\right)\partial_{x_{t_0}}p\left(x_{t_1} \mid x_{t_0}\right) + \frac{1}{2}g^2\left(x_{t_0}\right)\partial_{x_{t_0}}^2 p\left(x_{t_1} \mid x_{t_0}\right) \tag{13}$$

where $x_{t_0}$ and $x_{t_1}$ are distributions at the respective time steps. Specifically, it models how the distribution dynamics at a later point $t_1$ in time changes as we change $t_0$ at an earlier time.

In our case, we consider the diffusion over structure $\mathbf{y}_s$ and properties $\{\mathbf{y}_1, \ldots, \mathbf{y}_k\}$. The KFE of the system $\mathbf{y} = \{\mathbf{y}_s, \mathbf{y}_1, \ldots, \mathbf{y}_k\}$ is given by,

$$\partial_t p\left(\mathbf{y}_t\right) = -\partial_{\mathbf{y}_t}\left[f\left(\mathbf{y}_t\right)p\left(\mathbf{y}_t\right)\right] + \frac{1}{2}\partial_{\mathbf{y}_t}^2\left[g^2\left(\mathbf{y}_t\right)p\left(\mathbf{y}_t\right)\right] \tag{14}$$

**Independence Factorization**    We can factorize $p\left(\mathbf{y}_t\right)$ based on our assumption that the properties $\{\mathbf{y}_{1,t}, \ldots, \mathbf{y}_{k,t}\}$ are independent conditioned on the structure $\mathbf{y}_{s,t}$ as

$$\begin{aligned} p(\mathbf{y}_t) &= p(\mathbf{y}_{s,t}, \mathbf{y}_{1,t}, \ldots, \mathbf{y}_{k,t}) \\ &= p(\mathbf{y}_{s,t})p(\mathbf{y}_{1,t}, \ldots, \mathbf{y}_{k,t} \mid \mathbf{y}_{s,t}) \\ &= p(\mathbf{y}_{s,t})\prod_i^k p(\mathbf{y}_{i,t} \mid \mathbf{y}_{s,t}) \end{aligned} \tag{15}$$

Leveraging this factorization, we can define a system of SDEs with KFEs for each variable, leading us to the SDE system defined in Eq. 1 and Eq. 2.

**Reverse SDE:**    In the reverse case, we aim to denoise the full vector $\mathbf{y} = \{\mathbf{y}_s, \mathbf{y}_1, \ldots, \mathbf{y}_k\}$ where $\mathbf{y}_s$ denotes the diffusion over structure and $\{\mathbf{y}_1, \ldots, \mathbf{y}_k\}$ over the $k$ properties via reverse SDE. Expressing in the form of Eq. 13, we note that for $t_1 \geq t_0$,

$$-\partial_t p\left(\mathbf{y}_{t_1} \mid \mathbf{y}_{t_0}\right) = f\left(\mathbf{y}_{t_0}\right)\partial_{\mathbf{y}_{t_0}}p\left(\mathbf{y}_{t_1} \mid \mathbf{y}_{t_0}\right) + \frac{1}{2}g^2\left(\mathbf{y}_{t_0}\right)\partial_{\mathbf{y}_{t_0}}^2 p\left(\mathbf{y}_{t_1} \mid \mathbf{y}_{t_0}\right) \tag{16}$$

Anderson (1982) defines a joint distribution over the time-ordered variables $\mathbf{y}_{t_1}$ and $\mathbf{y}_{t_0}$ to derive the reverse SDE. We utilize their analysis and define a joint distribution

$$\begin{aligned} p\left(\mathbf{y}_{t_1}, \mathbf{y}_{t_0}\right) &:= p\left(\mathbf{y}_{s,t_1}, \mathbf{y}_{1,t_1}, \ldots, \mathbf{y}_{k,t_1}, \mathbf{y}_{s,t_0}, \mathbf{y}_{1,t_0}, \ldots, \mathbf{y}_{k,t_0}\right) \\ &= p\left(\mathbf{y}_{s,t_1}, \mathbf{y}_{1,t_1}, \ldots, \mathbf{y}_{k,t_1} \mid \mathbf{y}_{s,t_0}, \mathbf{y}_{1,t_0}, \ldots, \mathbf{y}_{k,t_0}\right)p\left(\mathbf{y}_{s,t_0}, \mathbf{y}_{1,t_0}, \ldots, \mathbf{y}_{k,t_0}\right) \end{aligned} \tag{17}$$

We denote $p(\mathbf{y}_{s,t_0}, \mathbf{y}_{1,t_0}, \ldots, \mathbf{y}_{k,t_0})$ by $p(\mathbf{y}_{t_0})$, and note that it can be decomposed similarly as in Eq. 15. Taking the time derivative of Eq. 17, we get

$$\begin{aligned} -\partial_t p\left(\mathbf{y}_{t_1}, \mathbf{y}_{t_0}\right) = &-\partial_t p\left(\mathbf{y}_{s,t_1}, \mathbf{y}_{1,t_1}, \ldots, \mathbf{y}_{k,t_1} \mid \mathbf{y}_{s,t_0}, \mathbf{y}_{1,t_0}, \ldots, \mathbf{y}_{k,t_0}\right)p(\mathbf{y}_{t_0}) \\ &- \partial_t p(\mathbf{y}_{t_0})p\left(\mathbf{y}_{s,t_1}, \mathbf{y}_{1,t_1}, \ldots, \mathbf{y}_{k,t_1} \mid \mathbf{y}_{s,t_0}, \mathbf{y}_{1,t_0}, \ldots, \mathbf{y}_{k,t_0}\right) \end{aligned} \tag{18}$$

**Comparison with KFE/KBE**  We observe that $\partial_t p\left(\mathbf{y}_{s,t_1}, \mathbf{y}_{1,t_1}, \ldots, \mathbf{y}_{k,t_1} \mid \mathbf{y}_{s,t_0}, \mathbf{y}_{1,t_0}, \ldots, \mathbf{y}_{k,t_0}\right)$ corresponds to the KBE in Eq. 16 and $\partial_t p(\mathbf{y}_{t_0})$ to the KFE in Eq. 14. Denoting $\{\mathbf{y}_{s,t_1}, \mathbf{y}_{1,t_1}, \ldots, \mathbf{y}_{k_1 1}\}$ by $\mathbf{y}_{t_1}$, we immediately get

$$
-\partial_t p\left(\mathbf{y}_{t_1} \mid \mathbf{y}_{t_0}\right) p(\mathbf{y}_{t_0}) - \partial_t p(\mathbf{y}_{t_0}) p\left(\mathbf{y}_{t_1} \mid \mathbf{y}_{t_0}\right)
$$

$$
= \left(f\left(\mathbf{y}_{t_0}\right) \partial_{\mathbf{y}_{t_0}} p\left(\mathbf{y}_{t_1} \mid \mathbf{y}_{t_0}\right) + \frac{1}{2} g^2\left(\mathbf{y}_{t_0}\right) \partial_{\mathbf{y}_{t_0}}^2 p\left(\mathbf{y}_{t_1} \mid \mathbf{y}_{t_0}\right)\right) p(\mathbf{y}_{t_0})
$$

$$
+ p\left(\mathbf{y}_{t_1} \mid \mathbf{y}_{t_0}\right)\left(\partial_{\mathbf{y}_{t_0}}\left[f\left(\mathbf{y}_{t_0}\right) p\left(\mathbf{y}_{t_0}\right)\right] - \frac{1}{2} \partial_{\mathbf{y}_{t_0}}^2\left[g^2\left(\mathbf{y}_{t_0}\right) p\left(\mathbf{y}_{t_0}\right)\right]\right)
$$

(19)

The derivatives can be handled, by following standard differentiation rules as,

$$
\partial_{\mathbf{y}_{t_0}} p\left(\mathbf{y}_{t_1} \mid \mathbf{y}_{t_0}\right) = \partial_{\mathbf{y}_{t_0}}\left[\frac{p\left(\mathbf{y}_{t_1}, \mathbf{y}_{t_0}\right)}{p\left(\mathbf{y}_{t_0}\right)}\right]
$$

$$
= \frac{\partial_{\mathbf{y}_{t_0}} p\left(\mathbf{y}_{t_1}, \mathbf{y}_{t_0}\right)}{p\left(\mathbf{y}_{t_0}\right)} - \frac{p\left(\mathbf{y}_{t_1}, \mathbf{y}_{t_0}\right) \partial_{\mathbf{y}_{t_0}} p\left(\mathbf{y}_{t_0}\right)}{p^2\left(\mathbf{y}_{t_0}\right)}
$$

(20)

Evaluating the derivative of the products in the forward Kolmogorov equation and substituting the derivatives accordingly we obtain,

$$
-\partial_t p\left(\mathbf{y}_{t_1}, \mathbf{y}_{t_0}\right) = \partial_{\mathbf{y}_{t_0}}\left[f\left(\mathbf{y}_{t_0}\right) p\left(\mathbf{y}_{t_0}, \mathbf{y}_{t_1}\right)\right] + \frac{1}{2} g^2\left(\mathbf{y}_{t_0}\right) \partial_{\mathbf{y}_{t_0}}^2 p\left(\mathbf{y}_{t_1} \mid \mathbf{y}_{t_0}\right) p(\mathbf{y}_{t_0})
$$

$$
- \frac{1}{2} p\left(\mathbf{y}_{t_1} \mid \mathbf{y}_{t_0}\right) \partial_{\mathbf{y}_{t_0}}^2\left[g^2\left(\mathbf{y}_{t_0}\right) p(\mathbf{y}_{t_0})\right]
$$

(21)

Matching the terms of the second-order derivatives with the expansion of the derivative and doing some algebraic manipulations, we obtain

$$
-\partial_t p\left(\mathbf{y}_{t_1}, \mathbf{y}_{t_0}\right) = \partial_{\mathbf{y}_{t_0}}\left[f\left(\mathbf{y}_{t_0}\right) p\left(\mathbf{y}_{t_0}, \mathbf{y}_{t_1}\right)\right] + \frac{1}{2} \partial_{\mathbf{y}_{t_0}}^2\left[p\left(\mathbf{y}_{t_1}, \mathbf{y}_{t_0}\right) g^2\left(\mathbf{y}_{t_0}\right)\right]
$$

$$
- \partial_{\mathbf{y}_{t_0}}\left[p\left(\mathbf{y}_{t_1} \mid \mathbf{y}_{t_0}\right) \partial_{\mathbf{y}_{t_0}}\left[g^2\left(\mathbf{y}_{t_0}\right) p\left(\mathbf{y}_{t_0}\right)\right]\right] ,
$$

(22)

which can be written as

$$
-\partial_t p\left(\mathbf{y}_{t_1}, \mathbf{y}_{t_0}\right) = -\partial_{\mathbf{y}_{t_0}}\left[p\left(\mathbf{y}_{t_1}, \mathbf{y}_{t_0}\right)\left(-f\left(\mathbf{y}_{t_0}\right) + \frac{1}{p\left(\mathbf{y}_{t_0}\right)} \partial_{\mathbf{y}_{t_0}}\left(g^2\left(\mathbf{y}_{t_0}\right) p\left(\mathbf{y}_{t_0}\right)\right)\right)\right] +
$$

(23)

$$
\frac{1}{2} \partial_{\mathbf{y}_{t_0}}^2\left[p\left(\mathbf{y}_{t_1}, \mathbf{y}_{t_0}\right) g^2\left(\mathbf{y}_{t_0}\right)\right]
$$

(24)

**Comparison with KFE**  The above result is in the form of a Kolmogorov forward equation with the joint probability distribution $p\left(\mathbf{y}_{t_1}, \mathbf{y}_{t_0}\right)$. The time-ordering is $t_1 > t_0$ and the term $-\partial_t p\left(\mathbf{y}_{t_1}, \mathbf{y}_{t_0}\right)$ describes the change of probability distribution as we move backward in time. We can marginalize over $t_1$, using the Leibniz rule, to obtain

$$
-\partial_t p\left(\mathbf{y}_{t_0}\right) = -\partial_{\mathbf{y}_{t_0}}\left[p\left(\mathbf{y}_{t_0}\right)\left(-f\left(\mathbf{y}_{t_0}\right) + \frac{1}{p\left(\mathbf{y}_{t_0}\right)} \partial_{\mathbf{y}_{t_0}}\left(g^2\left(\mathbf{y}_{t_0}\right) p\left(\mathbf{y}_{t_0}\right)\right)\right)\right] + \frac{1}{2} \partial_{\mathbf{y}_{t_0}}^2\left[p\left(\mathbf{y}_{t_0}\right) g^2\left(\mathbf{y}_{t_0}\right)\right]
$$

(25)

This finally gives a stochastic differential equation analogous to the Fokker-Planck/forward Kolmogorov equation that can be solved backward in time:

$$
d\mathbf{y}_{t_0} = \left(-f(\mathbf{y}_{t_0}, t) + \frac{1}{p\left(\mathbf{y}_{t_0}\right)} \partial_{\mathbf{y}_{t_0}}\left(g^2\left(\mathbf{y}_{t_0}\right) p\left(\mathbf{y}_{t_0}\right)\right)\right) dt + g\left(\mathbf{y}_{t_0}\right) d\mathbf{w}
$$

(26)

We keep $g^2\left(\mathbf{y}_{t_0}\right)$ independent of $\mathbf{y}_{t_0}$. Applying the log-derivative trick, the SDE simplifies to

$$
d\mathbf{y}_{t_0} = (f(\mathbf{y}_{t_0}, t) - g_{t_0}^2 \nabla_{\mathbf{y}_{t_0}} \log p(\mathbf{y}_{t_0})) dt + g_{t_0} d\mathbf{w}
$$

(27)

## A.2  CONDITIONAL SCORE FACTORIZATION

We extend our method to incorporate an external context or conditional information for conditional generation, similar to classifier-based (Dhariwal & Nichol, 2021) and classifier-free (Ho & Salimans,

2022) guidance. Following similar notation, the reverse SDE, given an external context $\mathbf{y}_C$ can be written as Song et al. (2021)

$$d\mathbf{y}_t = [\mathbf{f}(\mathbf{y}_t, t) - g_t^2 \nabla_{\mathbf{y}_t} \log p_t(\mathbf{y}_t, \mathbf{y}_C)]dt + g_t d\bar{\mathbf{w}} \tag{28}$$

Here $\mathbf{y}_t = \{\mathbf{y}_{s,t}, \mathbf{y}_{1,t}, \ldots, \mathbf{y}_{k,t}\}$, and $\mathbf{y}_C = \{\mathbf{y}_c \mid c \in C\}$ is an external context or conditioning variable. This external context can be a scalar or vector describing a property value of the primary variable like QED or plogp in the case of molecules or image labels in the case of images. The $\nabla_{\mathbf{y}_t} \log p_t(\mathbf{y}_t, \mathbf{y}_C)$ term pertains to the score function which guides the process (see table 2 for comparison with both classifier-based and classifier-free guidance). Under our condition independence assumption, the score function factorizes as

$$p_t(\mathbf{y}_{s,t}, \mathbf{y}_{1,t}, \ldots, \mathbf{y}_{k,t}, \mathbf{y}_C) = \prod_i^k p_t(\mathbf{y}_{i,t} \mid \mathbf{y}_{s,t}, \mathbf{y}_c) p_t(\mathbf{y}_{s,t}, \mathbf{y}_C) \tag{29}$$

$$\nabla_{\mathbf{y}_t} \log p_t(\mathbf{y}_{s,t}, \mathbf{y}_{1,t}, \ldots, \mathbf{y}_{k,t}, \mathbf{y}_C) = \nabla_{\mathbf{y}_t} \log p_t(\mathbf{y}_{s,t}, \mathbf{y}_C) + \sum_{i \notin C}^k \nabla_{\mathbf{y}_t} \log p_t(\mathbf{y}_{i,t} \mid \mathbf{y}_{s,t})$$
$$+ \sum_c^C \sum_i^k \delta_{i=c} \nabla_{\mathbf{y}_t} \log p_t(\mathbf{y}_{i,t} \mid \mathbf{y}_{s,t}, \mathbf{y}_c) \tag{30}$$

# B   DETAILS OF EXPERIMENTS

## B.1   CODE REPOSITORY

We provide the link to an anonymized repository for the computational experiments: `https://anonymous.4open.science/r/Forked-Diffusion-on-Graphs-F852`.

## B.2   CHEMICAL PROPERTIES EXPERIMENT

Further details for generation conditioned on chemical properties from Section 4.1.

**Datasets setup**   In terms of datasets, we train our models on the QM9 and ZINC250k datasets, and follow the preprocessing and the train/test splits outlined in (Jo et al., 2022). We consider 3 chemical properties for conditioning, namely: Penalized logP (plogP), Quantitative Estimate of drug-likeliness (QED), and SA score, which can be obtained given a molecule through the RDKit library (Landrum et al., 2016).

**Score models parameterization**   We follow (Jo et al., 2022) in terms of score-networks blocks and molecule representation, but adding the necessary modules for modeling the forked diffusion. The child process models properties $\mathbf{y}_i$ as follows:

$$\text{Pool}(\text{MLP}_i(\text{GNN}((\boldsymbol{X} \parallel \mathbf{y}_i, \parallel \mathbf{y}_c), \boldsymbol{A}))) \tag{31}$$

while the parent process also combines the contributions from properties $\mathbf{y}_i$ and the structure $\mathbf{y}_s$, as follows:

$$\text{MLP}(\text{GNN}(\boldsymbol{X}, \parallel \mathbf{y}_c, \boldsymbol{A})) + \sum_i \text{MLP}_i(\text{GNN}((\boldsymbol{X} \parallel \mathbf{y}_i \parallel \mathbf{y}_c), \boldsymbol{A})) \tag{32}$$

**Forking configuration**   In this experiment the forking diffusion is configured with a parent process that models the molecular graph information, and 3 child-processes (one for each considered property) to model the information pertaining to the chemical properties. The parent score network is trained to generate a graph, while the child score networks are trained to generate a property vector, via graph pooling operations. Parent and child processes are trained to evolve together as in Algorithm 1, and during generation, the output of the child process is reused into the parent process (Algorithm 2).

## B.3   QUANTUM PROPERTIES EXPERIMENT

Further details for generation conditioned on quantum properties from Section 4.2.

**Dataset setup**   The experiment is on QM9 dataset (Ramakrishnan et al., 2014). To ensure consistency and comparability with previous works (Hoogeboom et al., 2022; Bao et al., 2023; Huang et al., 2023), we adhere to the identical dataset preprocessing and training/test data partitions outlined by Huang et al. (2023).

**Forking configuration**   The parent process learns the score for the graph structure $\mathbf{y}_s$, and a child process learns the evolving property $\mathbf{y}_i$, corresponding to the property that we are conditioning on. The score networks corresponding to parent and child processes evolve both structure and labels as in Algorithm 1. Specifically, the parent network performs a diffusion operation, while the child network performs a diffusion operation and subsequently a classification. Note that differently from our previous experiment in 4.1, we use a single child process, for computational reasons.

**Score models parameterization**   We design modifications of the Diffusion Graph Transformer (DGT) (Huang et al., 2023). Within a DGT framework, the parent process model combines the evolving properties $\mathbf{y}_i$ and it combines it with the context $\mathbf{y}_c$, as:

$$\text{MLP}(\text{MLP}(\mathbf{y}_i) + \text{MLP}(\mathbf{y}_c)) \tag{33}$$

The child process network models the evolving properties $\mathbf{y}_i$ by performing a pooling operation after DGT diffusion, more specifically:

$$\text{Pool}(\text{DGT}(\mathbf{y}_s, \mathbf{y}_i, \mathbf{y}_c)) \tag{34}$$

### B.4 GENERIC GRAPH EXPERIMENT

Further details for generation conditioned on generic graph properties, from Section 4.3.

**Datasets setup**   We consider the following datasets: (1) Ego-small: 200 small ego graphs which are drawn from a larger Citeseer network dataset (Sen et al., 2008) and (2) Enzymes: Real protein graphs representing the tertiary structures of the enzymes from the BRENDA database (Schomburg et al., 2004). Properties:

1. # Nodes,
2. # Edges;
3. Density: the ratio of the number of nodes to the number of edges
4. ANND: Average Nearest Neighbor Degree

The properties can be extracted via network-X library (Hagberg et al., 2008).

**Model setup**   We employ a single child process for simultaneously evolving structure and one of the aforementioned properties. We use a parameterization based on (Jo et al., 2022), which we described in the experiment for chemical properties B.2. We consider our previous conditional GDSS baseline from the same experiment.

## C   EXPANDED RELATED WORKS

Digress (Vignac et al., 2023) proposes a classifier guidance approach based on DDPM (Ho et al., 2020), as a result, differs from continuous methodologies. MiCAM (Geng et al., 2023) generates goal oriented molecules on mined connection-aware motifs.

## D   EVALUATION METRICS

### D.1   MOLECULE GENERATION

When evaluating generative models for molecular graphs, domain-specific metrics are often employed to assess the quality of generated chemical structures. Used in chemical properties experiment 4.1 and for unconditional generation 4.4.

We follow metrics for evaluating molecules have been presented in previous works (Gómez-Bombarelli et al., 2018; Jin et al., 2018), that we describe as follows.

- **Validity (without correction)** is the fraction of valid molecules without valency correction or edge resampling, as done in (Jo et al., 2022).
- **Uniqueness**: Quantifies the diversity of generated molecules by computing the percentage of unique molecular graphs.
- **Novelty**: Evaluates the proportion of generated molecules that are dissimilar from molecules in the training dataset.
- **Frechet ChemNet Distance (FCD)** measures the distance between the test set and the generated set with the activation of the penultimate layer of ChemNet. Lower FCD values indicate more similarity between the two distributions.

### D.2 GRAPH GENERATION

Evaluation of graph generative models is a crucial aspect to assess their performance and quality in capturing the underlying structure of complex graph-structured data. Various evaluation metrics have been proposed to measure the fidelity, diversity, and other characteristics of generated graphs. Here we describe: 1) Maximum Mean Discrepancy (MMD) and 2) Degree, clustering, and orbit statistics, which are derived from MMD and were introduced from (You et al., 2018).

**Maximum Mean Discrepancy (MMD)** We use MMD for evaluating generic graph generation 4.3 and chemical properties conditional molecule generation 4.1.

MMD (Gretton et al., 2012) is a metric commonly used to quantify the statistical difference between distributions. Given two sets of samples, $X$ and $Y$, MMD computes the distance between their empirical means in a reproducing kernel Hilbert space (RKHS):

$$\text{MMD}^2(X, Y) = \frac{1}{n_X^2} \sum_{i,j=1}^{n_X} k(x_i, x_j) + \frac{1}{n_Y^2} \sum_{i,j=1}^{n_Y} k(y_i, y_j) - \frac{2}{n_X n_Y} \sum_{i=1}^{n_X} \sum_{j=1}^{n_Y} k(x_i, y_j), \quad (35)$$

where $k(\cdot, \cdot)$ is a positive definite kernel function and $n_X$, $n_Y$ are the sample sizes of $X$ and $Y$ respectively. MMD can be used to compare the distribution of real and generated graphs, providing insights into how well the model captures the true underlying data distribution.

**Degree, clustering, orbit** In the generic graph generation experiment 4.3 we use the metrics MMD statistics described here, which have been introduced by (You et al., 2018) and used in related works (Jo et al., 2022).

- **Degree Distribution**: The degree of a node in a graph is the number of edges connected to that node. The degree distribution, represented as a histogram, illustrates the frequency of nodes with different degrees. In the evaluation of graph generation models, it is crucial to assess how well the generated graphs replicate the degree distribution of real-world networks. Deviations from the expected degree distribution may indicate structural issues in the generated graph.
- **Clustering Coefficient**: The clustering coefficient of a node in a graph measures the extent to which its neighbors are also connected to each other, indicating the local connectivity or cliquishness of nodes. Evaluating a graph generation model involves comparing the clustering coefficients of generated graphs to those of real-world networks. A high-quality model should produce graphs with clustering coefficients similar to those observed in real networks, facilitating an evaluation of the local structure.
- **Orbit Statistics**: Orbit statistics involve counting different substructures in a graph, such as triangles and squares, and categorizing nodes into equivalence classes based on their roles in the network's structure. In evaluating graph generation models, we employ orbit statistics to capture global structural properties. By comparing the orbit statistics of

generated graphs to those of real-world networks, the quality of the generated structural features can be assessed. In our experiment, we follow previous works (You et al., 2018; Jo et al., 2022), and count the number of occurrences of all orbits with 4 nodes (to capture higher-level motifs).

