# OpenReview forum: "Forked Diffusion for Conditional Graph Generation"
_ICLR.cc/2024/Conference — Submitted to ICLR 2024_

### Official Review · Reviewer_hTgA · 2023-10-27

**Soundness:** 2 fair
**Presentation:** 1 poor
**Contribution:** 2 fair
**Rating:** 3
**Confidence:** 4

**Summary:**

This paper proposes the forked diffusion process, featuring one parent process on graph structures and children processes for distinct properties. The method is assessed using the QM9 and ZINC250K datasets in comparison to baselines. While the concept sounds interesting, the current study does not support the authors' conclusion (quoted below) on the last page.

> Specifically, FDP demonstrates exceptional capabilities in the domains of conditional graph generation for molecular structures, inverse
molecule design tasks, and the generation of generic graphs, surpassing contemporary diffusionbased methods.
>

**Strengths:**

1. The idea of incorporating forking into the diffusion process sounds reasonable.

2. The algorithm presented on page 6 is concise and aids in understanding the method.

**Weaknesses:**

W1: The paper lacks specific details on how the diffusion processes operate on graph structures and node features. Figure 1 illustrates the 3-D molecular structure; however, it remains unclear how, or if, these 3-D positions are integrated into the study.

W2: Some claims regarding contributions appear overstated. Recent research [1,2] indicates that achieving high levels of novelty, uniqueness, and validity on QM9 and ZINC250K is not particularly challenging; for instance, merely adding carbons can suffice. It's evident from recent studies [1,2] that both the genetic algorithm [3] and the reinvent method [5] serve as potent baselines. This paper should benchmark against a broader set of baselines to truly showcase its performance in molecular design tasks. Furthermore, molecular properties such as plogp and qed are often deemed impractical [1] and not truly beneficial for real-world inverse molecule design tasks.

W3: In GDSS, node features and graph structures are treated as separate diffusion processes. The connection between this approach and the proposed method requires deeper exploration.

W4: The assumption that "the properties are independent conditioned on the structure" in Equation 15 is unjustified and not reliable.

Other minor points:

W5: The pictures should be of higher resolution.

W6: There are several typos, such as the "}" symbol, in Figure 1.

W7: Definitions of the variables should be clarified. For instance, the dimension of the variable y is not clearly stated.

Ref.

[1] Sample Efficiency Matters: A Benchmark for Practical Molecular Optimization. NeurIPS 2022.

[2] Genetic algorithms are strong baselines for molecule generation. Arxiv.

[3] A graph-based genetic algorithm and generative model/monte carlo tree search for the exploration of chemical space. Chemical science.

[4] Molecular de-novo design through deep reinforcement learning. Journal of cheminformatics,

**Questions:**

1. Is it possible to include more baselines like the DiGress [1] for comparisons?

2. Some molecular structures in Figure 2 do not appear reasonable. Could the authors provide further analysis?

Ref.

[1] DiGress: Discrete Denoising diffusion for graph generation. ICLR 2023.

---

> ### Author Response · Authors · 2023-11-21
>
> Dear reviewer, thanks for raising thoughtful questions and valid points that help us to write better work. We address your points as follows.
>
> W1:
> The 3D information is incorporated into our experiment, as outlined in Section 4.2. Our architecture is derived from Jodo Huang et al. 2023 [4], specifically adapted to support the forking diffusion mechanism. Detailed information about the architecture is provided in Appendix B.3. We can expand this section if you believe this is necessary.
>
> For the experiment detailed in Section 4.1, we follow the setup of GDSS [4] (Jo et al 2022), utilizing only 2D information. Further specifics on this setup can be found in Appendix B.2.
>
> W2:
> We focus on property-conditional generation. We take into account all the best performing previous work considering the mentioned task that appeared into top conferences [1,2,3].
>
> Our goal is demonstrating the effectiveness of the learning mechanism of forked diffusion for this specific task. As a result, we follow the setup of established works on property conditional generation of molecules [1,2,3] .
>
>
>
> W3:
> Our method differs from GDSS as we are learning a joint dynamics between properties and structures. GDSS does not model properties. Differently from GDSS, our model is specifically designed for conditional tasks.
>
> W4:
> Our modeling approach adopts the assumption of independence among child properties in the forking process, a choice made for effective formulation. Whether chemical properties of molecules are considered independent or dependent depends on the specific characteristics under examination. While certain chemical properties may exhibit relative independence, others manifest interdependence influenced by the underlying molecular structure. Leveraging this understanding as an inductive bias, we integrate it into our forked diffusion process. This incorporation provides a more nuanced control over the joint space of properties and molecules, allowing for a refined representation that captures both independent and interdependent relationships in the data.
>
>
>
> W5: we will update
> W6: We have fixed this.
> W7: We define the dimension that are specific to each experiment in the Appendices B. The variable $Y_s$ represent adjacency matrix and node features, while $y_i$ is a scalar representing a graph property (QED, plogp, SAS) in case of molecules.
>
>
> Q1:
> We added 3 baselines: U-bounds, L-bounds, \#Atoms, following [1,2,3].
> The  upper bound ’U-bound’ baseline  randomly shuffles the property labels of molecules and reports their MAE. The ‘\#Atoms’ baseline predicts molecular properties based only on the number of atoms in molecules. Lower bound (L-bound) is the best performance bound from QM9.
>
>
> Digress comparison is not appropriate as its conditional generation task, because
> 1) Digress does not compare to any previous works on conditional generation, such as EDM [1], EEGSDE [2], Jodo [3], all these methods provide comprehensive experimental details, and we compare with them.
> 2) Digress does custom dataset processing, differently from the other benchmarks [1,2,3].
>
> In our first experiment (Section 4.1) we follow the setup of GDSS [4], in Section 4.2 we follow the setup from EDM [1], EEGSDE[3], Jodo[3].
>
> In addition, Digress codes presents reproducibility issues for the conditional generation, as mentioned by the authors on github: "Warning: The paper experiments were run with an old version of the code. We have incorporated the changes into the public version to create this branch, but we have not tested thoroughly yet. Please tell us if you find any bugs." The problem is confirmed by various issues raised on github \url{https://github.com/cvignac/DiGress/issues/68}.
>
> This said, we can still try if the reviewer think necessary, and we added digress to related work.
>
> Q2: Could you elaborate please? According to our understanding, Figure 2 presents valid molecular structures.
>
>
> Please let us know  whether we answered your points fairly, please consider to raise the score, in case we have, or let us know the issues that need to be fixed to improve the score and to improve the work.
>
> Best regards.
>
>
>
>
> References
>
> [1] Hoogeboom et al, "Equivariant diffusion for molecule generation in 3d", ICML 2022.
>
> [2] Bao et al, "Equivariant energy-guided sde for inverse molecular design", ICLR 2023.
>
> [3] H Huang "Learning Joint 2D \& 3D Diffusion Models for Complete Molecule Generation" 2023
>
> [4] Jo et al, "Score-based Generative Modeling of Graphs via the System of Stochastic Differential Equations" ICML 2022

---

> > ### Comment · Reviewer_hTgA · 2023-11-22
> > **Thank for the response**
> >
> > Thank the authors for the point-by-point response. I have read through them and found that some of my questions have not been clarified. For example, I did not find in the main text (including section 4.2) any explanation on how to use the atom coordinates in the generation model. I do not think the response to weakness 2 answers my question. Regarding Q2: a valid structure does not necessarily mean a useful structure. When looking at the example presented in Figure 2, the very large rings and weird molecule connections, to me, do not illustrate the advantages of the proposed method. They just mean that the generated example is not realistic, hardly synthesized, and of limited use. Overall, I am inclined to keep my score.

---

### Official Review · Reviewer_qecv · 2023-10-29

**Soundness:** 3 good
**Presentation:** 2 fair
**Contribution:** 2 fair
**Rating:** 5
**Confidence:** 3

**Summary:**

This work presents conditional diffusion framework for graph generation by proposing forked diffusion processes, that models the graph diffusion process as a single parent process over a primary variable (i.e., structure) and multiple child processes over dependent variables, further including additional context. This work provide experimental results on diverse graph generation tasks showing improved generation performance over continuous graph diffusion model GDSS and other generative models.

**Strengths:**

- This work propose new graph diffusion framework by modeling a system of joint diffusion processes with explicit dependency conditions, e.g., dependency of child variables on parent variable, and additional context.

- The experimental results on diverse tasks demonstrate that FDP improves generation performance over other graph diffusion models, especially GDSS.

**Weaknesses:**

- Explanation on key components lacks details:
  1. What do the structure variable ($x_s$) and child variables ($x_p$) actually represent during the experiments?
  2. What does the models ($s_{\\phi_i, t}$) approximate?
  3. How is the training objective derived? Is it a straightforward extension of score matching objective?

- Several claims made in the paper are not clear and requires justification:
  1. ( as explained in Intro) How does the parent process guide the childe processes? From what I have understood, the paraent process and child processes are dependent to each other rather than specific process guiding others, similar to the processes of GDSS. In particular, the proposed diffusion framework seems to be a system of multiple processes (parent and childs) with some dependency conditions given betwen the variables, and the score functions are dependent to all the variables including additional context.
  2. Is the assumption on independence between child variables valid for the experiments? As what I have understood, the child variables represent some chemical properties, for which do not seem to be independent to each other.
  3. (at the end of Sec. 3.4 and Tab. 1) What does the energy guidance mechanism indicate? Why is the additional influence of $s_{\\phi_i}$ characterized as an energy guidance?

- As FDP requires multiple score models (for the primary and child variables), I presume the number of model parameters required  for FDP would be quire large. Ablation study on number of model parameters could strengthen the effectiveness of the proposed framework.

- Important experimental details, for example, model architecture or what the variables actually represents, are not provided in the main paper. There seems to be some explanation in Appendix B, but not referenced by the main paper.

- The reason for superior performance on unconditional molecule generation (Sec. 4.4) is not clear. Especially, an important baseline, GDSS, seems to be missing in Tab. 6.

- Missed some previous works on conditional graph generation:
  - Vignac et al., DiGress: Discrete Denoising diffusion for graph generation, ICLR 2023
  - Lee et al., Exploring Chemical Space with Score-based Out-of-distribution Generation, ICML 2023

**Questions:**

- Please address the questions in the Weakeness section.

- Is the Wiener processes ($\\mathrm{d}\\textbf{w}$) in Eq. (2) independent?

- Recent works (e.g., DiGress) find that using Graph Transformer architecture (instead of GNN-based architecture of GDSS) shows improved generation performance. Does FDP show similar improvement over GDSS when using the Transfomer architecture?

---

> ### Author Response · Authors · 2023-11-21
>
> Dear reviewer, thanks for helping us improving our work. We address your points as follows.
>
> W1.1:
> The parent and child variables in each experiment vary, typically with the structure variable representing the graph and the child variables representing associated properties. Appendices B2, B3, and B4 provide detailed explanations for these variables.
>
> W1.2: $S_{\phi_i, t}$ serves as an approximation for the coupling between the $i^{th}$ child variable (or property) and the parent variable (graph). This influence on the reverse Stochastic Differential Equation (SDE) process enhances control over the conditional generation, allowing for a more nuanced and precise generation process.
>
> W1.3: The training objective is based on score-matching but incorporates the forked diffusion settings given by propositions 1 and 2, which are novel results that we obtained via formal derivations in Appendix A.1 and A.2.
>
> W2.1: A singular parent diffusion process is linked to a primary variable, such as the graph structure, while multiple child diffusion processes are dedicated to dependent variables, such as graph properties or labels. The parent process orchestrates the co-evolution of its child processes, as mathematically depicted by the system of equations governing the forward Stochastic Differential Equation (SDE) process in Eq. 2. Here, $y_{s}$ represents the parent variable (graph structure), and $y_{i}$ represents the dependent variable, i.e., properties. This influence persists in the reverse SDE, as illustrated in Eq. 4.
>
> W2.2: We adopt the assumption of independence among child properties as a modeling choice to formulate the forking process effectively. The consideration of chemical properties of molecules as either independent or dependent is contingent on the specific characteristics being examined. While certain chemical properties can be relatively independent, others exhibit interdependence, influenced by the underlying molecular structure. Leveraging this inductive bias, we incorporate this into our forked diffusion process to achieve a more nuanced control over the joint space of properties and molecules.
>
> W2.3: $S_{\phi_i, t}$ serves as an approximation for the coupling between the $i^{th}$ child variable (or property) and the parent variable (graph). See the answer at point W1.2.
>
> W4: Could you let us know what kind of ablation study you require ? Is it like using all (n out of n) properties as children vs 1 out of n properties etc?
>
> W5: We agree, will move such important details in the main paper.
>
> W6: We agree, will move the unconditional experiment into appendix, as our method is mostly focused on conditional generation.
>
> W7: Thanks for pointing out those relevant works. We will add into related work section.
>
> Q1: We generate unique noise samples for each child process to ensure the validity of the conditional assumption. The conditional information for each child process is derived from the diffusion function, which, in turn, is conditioned on the specified structure.
>
> Q2: Thanks for the question. In fact, our implementations for are using transformers. With a careful read of the GDSS work [1] one can see that they are employing attention mechanisms. In fact, the architecture for processing graphs in GDSS is the graph multi-head attention (Baek et al.,2021) [2] . Our method in experiment 4.1 is built using similar architectures. For the experiment in 4.2 we compare to Jodo [3], which also uses Transformers, and we inherit a similar architecture for the our forked implementations for Section 4.2.
>
>
> If the concerns you raised have been satisfactorily resolved, we kindly request you to consider raising the score. Your feedback is valuable to us, and we want to ensure your satisfaction. If there are any remaining issues or if you require further clarification, please do not hesitate to let us know. Best regards.
>
>
>
> References
>
> [1] Jo et al, "Score-based Generative Modeling of Graphs via the System of Stochastic Differential Equations" ICML 2022
>
> [2] Baek et al, "Accurate learning of graph representations with graph multiset pooling" ICLR 2021
>
> [3] H Huang "Learning Joint 2D \& 3D Diffusion Models for Complete Molecule Generation" 2023

---

### Official Review · Reviewer_FTyY · 2023-11-01

**Soundness:** 2 fair
**Presentation:** 2 fair
**Contribution:** 2 fair
**Rating:** 5
**Confidence:** 3

**Summary:**

This paper proposes a forked diffusion model for conditional graph generation that introduces parent process and child processes to learn and generate graphs with desired properties. The contributions of this paper include introducing forking as a new technique for conditional generation, providing a rigorous mathematical framework using SDE, and demonstrating the versatility of the proposed forked diffusion with empirical evidence.

**Strengths:**

1.	The forked diffusion framework for graph generation is novel.

**Weaknesses:**

1. The motivation behind the proposal of forked diffusion is not clear in the introduction. While the categories of related work are presented, it is unclear what deficiencies these methods have compared to the proposed approach.
2. The related work is not comprehensive enough, as recent diffusion-based methods such as Digress [1], etc., have not been mentioned.
3. Given the computational demands of the diffusion model, I believe that the application of the forked diffusion model to large-scale datasets may be even more challenging. The authors should compare the proposed model's training and generation efficiency with related work on large-scale datasets such as Guacamol.
4. In the conditional generation experiments, experiments on large-scale datasets and additional baselines for comparison should be included. Especially for the task of molecular generation, it would be beneficial for the authors to include recent methods such as DiGress [1], MiCaM [2], MolHF [3], etc.

[1] Vignac, Clement, et al., "DiGress: Discrete Denoising Diffusion for Graph Generation." The Eleventh International Conference on Learning Representations, 2023.

[2] Geng, Zijie, et al., "De Novo Molecular Generation via Connection-Aware Motif Mining." The Eleventh International Conference on Learning Representations, 2023.

[3] Zhu, Yiheng, et al., "MolHF: A Hierarchical Normalizing Flow for Molecular Graph Generation." International Joint Conference on Artificial Intelligence, 2023.

**Questions:**

1. It is unclear why the molecular metrics results for the Zinc dataset are not presented in Table 3.
2. It is unclear why there is no comparison of unique metrics in Table 6.

---

> ### Author Response · Authors · 2023-11-21
>
> Thanks for your useful and detailed review that improves our work. We address your points as follows:
>
> W1:
> We introduce an innovative method for refining guidance within diffusion models to achieve superior controllable generation. Our approach adopts a unique perspective by concurrently modeling the variable and its properties, leading to an aligned latent space. Empirically, our method outperforms naive conditional diffusion models guided by either classifier-free or classifier-based methods [5]. Furthermore, we establish a robust mathematical framework using Stochastic Differential Equations (SDEs) to derive both the forward and backward forking diffusion processes.
>
>
> W2:
> Thank you for the reference. We have added Digress in Appendix C (extended related works), and will further refine, to bring it in related works. We put this reference appropriately in our paper.
>
>
> W3:
> Thanks, we are in the process of developing the large scale experiment on Guacamol. However, we note that our work focuses on the task of property conditional generation, that has been published into ICML [4] and ICLR [5]. We compare to such works, following their dataset setup.
>
>
> W4:
> Thank you for providing the pertinent references for our research. We have incorporated these citations in appendix C. It's important to highlight that MolHF [3] does not possess a property conditional generation feature, thereby hindering a direct comparison with models that exhibit this capability.
>
> While the MiCAM work [2] is related (included in the Expanded Related Works section in appendix C), it does not evaluate on Mean Absolute Error (MAE) like the literature we are following for property-conditional generation (e.g., [4,5]). In MiCAM [2], the task is goal-directed generation, making it not directly comparable with any diffusion approach considered in [4,5].
>
>
> Q1: Thanks for pointing our this issue. We are running the ZINC experiment and will complete that evaluation.
>
> Q2: The Uniqueness results were equal to 1 for all methods, we will complete the table. Thanks for the help in the details.
>
> Final remark:
> We thank the reviewer for the pointers on improving the paper. If we have addressed the questions, please consider increasing the score, or let us know what other points are still missing/ unclear/ necessary for improving this work!
>
> Best regards.
>
>
> References
>
> [1] Vignac, Clement, et al., "DiGress: Discrete Denoising Diffusion for Graph Generation." The Eleventh International Conference on Learning Representations, 2023.
>
> [2] Geng, Zijie, et al., "De Novo Molecular Generation via Connection-Aware Motif Mining." The Eleventh International Conference on Learning Representations, 2023.
>
> [3] Zhu, Yiheng, et al., "MolHF: A Hierarchical Normalizing Flow for Molecular Graph Generation." International Joint Conference on Artificial Intelligence, 2023.
>
> [4] Hoogeboom et al, "Equivariant diffusion for molecule generation in 3d", ICML 2022.
>
> [5] Bao et al, "Equivariant energy-guided sde for inverse molecular design", ICLR 2023.

---

> > ### Comment · Reviewer_FTyY · 2023-11-22
> >
> > Thank you for your response. However, I believe that some necessary experimental results are still missing, and I will maintain my score.

---

### Official Review · Reviewer_GVsE · 2023-11-02

**Soundness:** 2 fair
**Presentation:** 3 good
**Contribution:** 1 poor
**Rating:** 3
**Confidence:** 4

**Summary:**

This paper works on conditional graph generation via score-based generative model. Instead of directly input conditional properties as context to model inside generation, the paper proposes to use a separate variable to model each conditional property and jointly model them inside the score-based diffusion framework. The author tested it over many real-world datasets, including molecular datasets and generic graphs. The proposed method shows better performance to naive conditional diffusion models.

**Strengths:**

1. The proposed method is easy-to-follow, and the written is clear.
2. While being a simple extension via introducing additional variables for conditional properties inside diffusion process, the method shows better performance comparing to use these properties as context input directly.
3. The author did a relatively comprehensive evaluation on many datasets and properties.

**Weaknesses:**

1. The proposed method is kind of a simple extension to current SDE diffusion models. There is not much technique improvement except introducing additional variables for these conditional properties inside the SDE process.
2. Most importantly, as the author just directly introducing these additional variables (with a simple conditional independent assumption among properties given the graph), the properties are not aligned with the intermediate graphs during the SDE process. This makes the method not very reasonable. For example, as there is no direct correspondance between graphs and properties, it is possible that the generated properties are not aligning with the generated graph. The alignment between property and graph at individual level is very important, all experiments don't have evaluation on this individual alignment. Only population-level MAE for these properties are evaluated.
3. In the experiment of generic graph generation, I don't see a clear difference of the population-level MAE between the proposed method and naive baseline GDSS, which may indicates that the method is not very effective.
4. Table 6 shows the result over unconditional generation, although this is not the intention of the designed method, I'm curious why the baseline only include GDSS-seq instead of the original GDSS.
5. Figure 3, ego-small should be the right one?

**Questions:**

1. Can you summarize what you want to prove inside Appendix A.2? I'm kind of confused as I believe the reverse SDE formulation is a well-known result that is directly used.

---

> ### Author Response · Authors · 2023-11-21
>
> Thanks for your thoughtful review which is helping to improve our paper, please see our replies below:
>
> W1:
>
> The novelty of our approach is given by the following point:
> We present an unexplored approach to characterize the propagation of conditional information, or guidance, within diffusion models, aiming for enhanced controllable generation. Our method introduces a novel perspective by jointly modeling the variable and its properties, resulting in an aligned latent space. Empirically, our approach outperforms naive conditional diffusion models, guided by classifier-free or classifier-based methods. Additionally, we establish a robust mathematical framework utilizing Stochastic Differential Equations (SDEs) to derive both the forward and backward forking diffusion processes.
>
>
> W2:
>
> Our objective is to capture the joint dynamics between the graph (parent variable) and its associated properties (child variables) through our forked diffusion processes, both in the forward and reverse directions. In the inference phase, the reverse Stochastic Differential Equation (SDE) responsible for graph generation is influenced by the properties and their reverse process dynamics (refer to Eq. 5). This nuanced interplay allows for superior control over the generation process, showcasing exceptional performance compared to conventional methods relying on classifier-free or classifier-based guidance. It's worth noting that, like many molecule generation and property prediction methods in the Deep Learning literature [1,2], usually population-level MAE is used as a metric to quantify the performance of the generated molecules via the model which may result in some discrepancy towards the alignment. We follow the setups from such previous works [1,2], which have been published in top conferences (ICML,ICLR) and highly cited.
>
>
>
>
>
>
> W3: In the context of generic graph generation, we extracted graph properties from the data using the NetworkX library and benchmarked our approach against GDSS [3], one of the most recent state-of-the-art methods in this domain. Notably, our method demonstrates superior performance compared to existing alternatives. While these properties may not be as inherently intuitive as molecular ones, they serve as a testament to the efficacy of our approach, particularly on synthetic datasets.
>
>
> W4: We agree that the unconditional tasks for our model did not show a strong result. Our method is mostly suitable for controlled generation, and will move this experiment into the appendix.
>
> W5: Yes, we will fix the typo.
>
> Q1: In Appendix A.2, we elucidate the reverse Stochastic Differential Equation (SDE) system governing our forked diffusion process. The derivation relies on the theoretical application of both forward and backward Kolmogorov equations to establish the reverse SDE for the joint system encompassing the graph and its associated properties.
>
> We are confident that our response effectively addresses your concerns. If we have successfully addressed your questions, we kindly ask you to consider raising your score. Alternatively, if there are areas where we can further improve to better meet your expectations, please let us know. Your feedback is highly valued, and we appreciate your cooperation. Thank you.
>
>
> References:
>
> [1] Hoogeboom et al, "Equivariant diffusion for molecule generation in 3d", ICML 2022.
>
> [2] Bao et al, "Equivariant energy-guided sde for inverse molecular design", ICLR 2023.
>
> [3] Jo et al, "Score-based Generative Modeling of Graphs via the System of Stochastic Differential Equations" ICML 2022

---

> > ### Comment · Reviewer_GVsE · 2023-11-22
> > **Response to authors' rebuttal**
> >
> > Thank you for your response. After reading, I am more confident that my first round evaluation is correct. Hence I would like to keep the score.

---

### Comment · Area_Chair_ENJa · 2023-11-22
**Less than one day remaining**

Dear Reviewers,

If you have already responded to authors last response, Thank you!
If not, please take some time, read their responses and acknowledge by replying to the comment. Please also update your score, if applicable.

Thanks everyone for a fruitful, constructive, and respectful review process.

Cheers, Your AC!

---

### Meta-Review · Area_Chair_ENJa · 2023-12-09

**Metareview:**

The paper proposes a conditional generative model with a parent diffusion process for the graph structure and child processes for propertis. The intention is allowing better control over conditional generation. The method is evaluated on tasks like molecular graph generation and the paper shows improved performance over baselines.

Strengths:
- A rigorous matematical framework based on stochastic differential equations.
- It demonstrates versatility through evaluations on multiple graph generation tasks.

Weaknesses:
- In problem formulation the motivation, assumptions,  and comparison to related works are a little unclear. It does not clearly highlight shortcomings of existing methods to contrast them to the current work. Some details on model architetures and training procedures are missing and can be expanded.
- In evaluations large-scale experiments, additional baselines and analyses could have been added for a stronger support. The evaluation on unconditional generation seems incomplete. The superior generative capability compared to baselines is not conclusively supported.
- As a minor point, there are issues with variable definitions and presentations.

The paper can be improved on the above 3 pillars.

**Justification For Why Not Higher Score:**

The paper at the current stage needs a major revision for being able to publish.

**Justification For Why Not Lower Score:**

N/A

---

### Decision · Program_Chairs · 2024-01-16

Reject